# OpenReview forum: "HoliSafe: Holistic Safety Benchmarking and Modeling for Vision-Language Model"
_ICLR.cc/2026/Conference — ICLR 2026 Conference Withdrawn Submission_

### Official Review · Reviewer_4KEi · 2025-10-27

**Soundness:** 3
**Presentation:** 2
**Contribution:** 3
**Rating:** 4
**Confidence:** 4

**Summary:**

The paper proposes HoliSafe, a holistic dataset and benchmark that systematically covers all five combinations of safe/unsafe image–text pairs, aiming to address incomplete coverage in prior VLM safety datasets. It also introduces a Visual Guard Module (VGM) that through training, enables VLMs to jointly classify image harmfulness and generate safe responses. Comprehensive experiments across open and closed VLMs show that models trained with HoliSafe and VGM achieve improved safety on existing and new benchmarks.

**Strengths:**

- The experiments, evaluated models, and datasets are comprehensive, reflecting substantial effort and careful execution. I appreciate the thoroughness of the evaluation and the significant work that clearly went into conducting it.

- The motivation to build a comprehensive benchmark that systematically covers all possible combinations of modality fusions is strong. To the best of my knowledge and based on the paper’s claims, prior works and benchmarks have not achieved this level of completeness or focus in addressing multimodal safety coverage.

**Weaknesses:**

- While the paper criticizes prior works for lacking architectural innovations to enhance safety, its own proposed Visual Guard Module (VGM) is conceptually simple; a small MLP added on top of the VLM, and thus does not constitute a substantial architectural contribution. I believe the contribution should still emphasize the benchmark and the data.

- Since the proposed approach updates not only the new module but also the vision encoder and adapter weights, unlike most standard post-training safety methods, it would have been more informative to report both refusal-rate (RR) and benign capability metrics separately and in greater detail across various helpfulness and capability benchmarks. This would clarify the trade-offs and potential utility degradation resulting from the added safety training and architectural revisions.

- The role of the Visual Guard Module (VGM) is unclear in cases the image is benign. If the VGM is trained solely to assess image harmfulness, it is uncertain how it behaves in such scenarios or what it actually predicts. Moreover, training a classifier to always assess the harmfulness of visual inputs may introduce bias, potentially degrading utility or altering model behavior even for benign, non-safety-related inputs.

- In response to the claims around lines 142–143, the paper overlooks relevant prior works that already consider similar safety combinations. For instance, the Multimodal Situational Safety (ICLR 2025) paper also includes the “Safe image + Safe text” configuration, and JailbreakV-28K (COLM 2024) addresses the “Unsafe image + Safe text” case. It is unclear why these studies were not cited or discussed.

- Since a substantial portion of the benchmark is generated using LLMs, it would be valuable to include a discussion on the diversity of the generated instructions and images. “Diversity” is a well-known limitation in LLM-based benchmark creation, and analyzing it would strengthen the dataset’s credibility. Additional embedding-based analyses or distribution visualizations could further help demonstrate the coverage and variety of the collected samples.

- Sections 2 and 3 are overly text-dense, which makes the reading experience a bit boring. I suggest revising these sections to improve readability and presentation; perhaps by reducing wordiness, summarizing high-level insights more concisely, and adding illustrative elements such as figures or diagrams to visualize the stages of benchmark construction. There are also several repetitive statements (e.g., repeatedly noting that *Unlike others, HoliSafe-Bench covers all possible combinations*. Overall, the presentation of the paper can be improved a lot.

**Questions:**

Please refer to the weaknesses. Thank you.

---

### Official Review · Reviewer_fJWv · 2025-10-30

**Soundness:** 2
**Presentation:** 3
**Contribution:** 3
**Rating:** 4
**Confidence:** 4

**Summary:**

This paper introduces HoliSafe, a safety-oriented dataset for vision-language models (VLMs) that includes both training and testing splits. The dataset covers a wide range of safety-related combinations across the two modalities. The authors further design VGM, a plug-in component for VLMs that determines whether an input image is safe. Extensive experiments demonstrate the challenges posed by HoliSafe-Bench and highlight the superiority of the fine-tuned model and VGM in both harmlessness and helpfulness evaluations.

**Strengths:**

1. The proposed dataset is comprehensive and contains various multimodal safety combinations.
2. The paper is well organized
3. The VSG design is novel and interpretable
4. The performance of the fine-tuned model is strong in both safety and utility tasks.

**Weaknesses:**

1. **The Reliability of Categorizing Images by Safety Category and Safeness**
   - The images are labeled by human experts and GPT-4o. However, the criteria for determining image safety are somewhat ambiguous. For example, I believe the second image in Figure 1 is safe, as it only depicts an ID card and keys. In contrast, the fourth image in Figure 1 conveys a sense of violent intent, which I consider unsafe.
2. **Lack Validation of GPT-4o Generated Instruction-Response Pairs**
   - The paper lacks validation of GPT-4o-generated instruction-response pairs, which may lead to low-quality generation or incorrect information in the generated data. Besides, the harmlessness of GPT-4o-generated training labels should be checked to ensure the effectiveness of training data.
3. **Inconsistency between Figure and Main Paper (Line 266)**
   - The image in Figure 2 is about property crimes. However, in line 266 of the paper, it claims the content as drug-related hazards.
4. **Evaluation on Vision Guard Models is Unfair**
   - The training and testing data in HoliSafe are quite similar, making it unfair to assess the vision guard model’s performance solely on the HoliSafe test set. The evaluation would be more convincing if additional vision-related benchmarks were included.
# Typos
Figure 3 Caption: mASR -> ASR

**Questions:**

See Weakness

---

### Official Review · Reviewer_ojhU · 2025-10-30

**Soundness:** 2
**Presentation:** 3
**Contribution:** 2
**Rating:** 2
**Confidence:** 4

**Summary:**

This paper proposes a safety benchmark for Vision-Language Model, namely HoliSafe. Compared with the existing benchmarks, this dataset is distinct since it defines five safe-unsafe image-text combinations. Furthermore, this paper proposes a visual guard module (VGM) to assess the harmfulness of input images for VLMs. Experiments show that Safe-VLM with VGM, trained on Holisafe, achieves the best safety performance across multiple VLM benchmarks.

**Strengths:**

1.	Regarding the safety benchmarks, this paper combines different images and texts together to make it contain a large variety of image-text combination types.

**Weaknesses:**

1. As we can see from Table 1, compared with the existing benchmarks, HoliSafe does not include new types of combinations. In other words, I am confused about the motivation of this work. Why can't we just merge all the existing benchmarks together to obtain a more comprehensive dataset?

2. Due to the ill-posed motivation, I am concerned about the significance of this work for both academia and industrial communities.

3. The VGM method is incremental. By including a classifier in the model to identify the safety category, the output will be generated based on the classification results.

**Questions:**

1.	See Weaknesses No.1.

2.	As in this paper, we have explicated the different combinations of image-text inputs, what insights can we get from these combinations? Or what can we learn from the benchmarks for future model development?

---

### Note · Authors · 2025-11-13

I have read and agree with the venue's withdrawal policy on behalf of myself and my co-authors.